# Learning from Mixtures of Private and Public Populations

**Raef Bassily**
Department of Computer Science & Engineering
The Ohio State University
bassily.1@osu.edu

**Shay Moran**
Department of Mathematics
Technion – Israel Institute of Technology
smoran@technion.ac.il

**Anupama Nandi**
Department of Computer Science & Engineering
The Ohio State University
nandi.10@osu.edu

## Abstract

We initiate the study of a new model of supervised learning under privacy constraints. Imagine a medical study where a dataset is sampled from a population of both healthy and unhealthy individuals. Suppose healthy individuals have no privacy concerns (in such case, we call their data "public") while the unhealthy individuals desire stringent privacy protection for their data. In this example, the population (data distribution) is a mixture of private (unhealthy) and public (healthy) sub-populations that could be very different.

Inspired by the above example, we consider a model in which the population $\mathcal{D}$ is a *mixture* of two sub-populations: a private sub-population $\mathcal{D}_{\mathsf{priv}}$ of private and sensitive data, and a public sub-population $\mathcal{D}_{\mathsf{pub}}$ of data with no privacy concerns. Each example drawn from $\mathcal{D}$ is assumed to contain a privacy-status bit that indicates whether the example is private or public. The goal is to design a learning algorithm that satisfies differential privacy only with respect to the private examples.

Prior works in this context assumed a homogeneous population where private and public data arise from the same distribution, and in particular designed solutions which exploit this assumption. We demonstrate how to circumvent this assumption by considering, as a case study, the problem of learning linear classifiers in $\mathbb{R}^d$. We show that in the case where the privacy status is correlated with the target label (as in the above example), linear classifiers in $\mathbb{R}^d$ can be learned, in the agnostic as well as the realizable setting, with sample complexity which is comparable to that of the classical (non-private) PAC-learning. It is known that this task is impossible if all the data is considered private.

## 1 Introduction

Despite the remarkable progress in privacy-preserving machine learning powered by the rigorous framework of differential privacy (DP) [DMNS06], the current state of the art has several limitations. Most of the existing works on differentially private learning follow a conventional model, where the entirety of the input dataset to the learning algorithm is assumed to be sensitive and private, and hence, requires protection via the stringent constraint of DP. Unfortunately, this conservative approach has fundamental limitations that manifest in many problems. For example, learning even simple classes of functions (e.g., one-dimensional thresholds over $\mathbb{R}$) is provably impossible under that stringent model [BNSV15, ALMM18] even though such classes are trivially learnable without privacy constraints.

More recent works [BNS13, BTT18, ABM19, NB20, BCM⁺20] have considered a more relaxed model, where the input dataset is made up of two parts: a private sample (as in the conventional model), and a "public" sample that entails no privacy constraints. In this model, the algorithm is required to satisfy DP only with respect to the private sample. Despite of the good news brought by these works showing the possibility of circumventing some of the aforementioned limitations by harnessing a limited amount of public data, *all these works make the strong assumption that both the private and public samples come from the same population (i.e., they arise from the same distribution).* This can limit the practical value of these results in many real-life scenarios, where private and public data are naturally distinct.

Indeed, for a data record, the attribute of being sensitive can be strongly correlated with the value of that record (i.e., the realization of the feature-vector and the label). For example, imagine a scenario where a bank wants to predict the credit-worthiness of applicants for a credit card. To do this, a training sample is drawn from a population of individuals with good and bad credit scores. Suppose individuals with good credit score have no privacy concerns in sharing their data with the bank (and hence, their data can be viewed as "public"), while those with bad credit score are concerned about what the study may reveal about them to third parties and understandably so, because they do not want such information to impact their chances in future opportunities. In this example, the population is a mixture of two very different groups: a sub-population with a good credit score (public sub-population), and a sub-population with bad credit score (private sub-population).

In this work, we introduce a new model for differentially private learning in which a learning algorithm has access to a mixed dataset of private and public examples that arise from possibly different distributions. The algorithm is required to satisfy DP only with respect to the private examples. More specifically, in our model, the underlying population (data distribution) $\mathcal{D}$ is a *mixture* of two possibly *different* sub-populations: a private sub-population $\mathcal{D}_{\mathsf{priv}}$ of sensitive data, and public sub-population $\mathcal{D}_{\mathsf{pub}}$ of data that is deemed by its original owner to have no risk to personal privacy. We assume that each example drawn from the mixture $\mathcal{D}$ has a "privacy flag" which is a binary label to indicate whether the example is private or public. As usual in the statistical learning framework, we do not assume the knowledge of $\mathcal{D}$ or any of the sub-populations (or their respective weights in the mixture).

**Contributions**

- **Introducing PPM model:** We formally describe the basic model of supervised learning from mixtures of private and public populations, and define the corresponding class of learning algorithms, which we refer to as *Private-Public Mixture (PPM)* learners.

- **Learning Halfspaces:** Although the first quick impression about the model might be that it is a bit too general to allow for interesting results beyond what is covered by the conventional model of DP learning, we demonstrate that this is not the case and prove the first non-trivial result under this model in the context of learning halfspaces (linear classifiers) in $\mathbb{R}^d$ (for any $d \geq 1$). We give a construction of a PPM learner for this problem in the case where the privacy status is correlated with the target label, as in the credit-worthiness example above. Curiously, our PPM learner is improper: it outputs a hypothesis (classifier) that can be described by at most $d$ halfspaces. We hence derive upper bounds on the sample complexity of this problem in both the realizable and agnostic settings. In particular, we show that halfspaces in $\mathbb{R}^d$ can be learned in the aforementioned PPM model up to (excess) error $\alpha$ using $\approx \frac{d^2}{\alpha}$ total examples in the realizable setting, and using $\approx \frac{d^2}{\alpha^2}$ total examples in the agnostic setting. As noted earlier, in the conventional model, where all the examples drawn from $\mathcal{D}$ are considered private, this class cannot be learned in $\mathbb{R}^d$ (even for $d = 1$) [BNSV15, ALMM18] [1]. Our bounds are comparable to the classical, non-private sample complexity of learning halfspaces. In particular, our bounds are only a factor of $d$ worse than their non-private counterparts. We leave the question of whether our bounds can be improved to future work.

**Techniques:** The idea of our construction for learning halfspaces goes as follows. First, we use the public examples to define a *finite* family of halfspaces $\widetilde{\mathcal{C}}_{\mathsf{pub}}$. Then, we employ a useful tool from

convex geometry known as Helly's Theorem [Hel23] to argue the existence of a collection of at most $d$ halfspaces in $\widetilde{\mathcal{C}}_{\mathsf{pub}}$ whose intersection is disjoint from the ERM halfspace (the halfspace with smallest empirical error with respect to the entire set of training examples). This implies that there is a hypothesis (described by the intersection of at most $d$ halfspaces from $\widetilde{\mathcal{C}}_{\mathsf{pub}}$) whose empirical error is not larger than that of the ERM halfspace. Hence, we reduce our learning task to DP learning of a *finite* class $\mathcal{G}$ that contains all possible intersections of at most $d$ halfspaces from $\widetilde{\mathcal{C}}_{\mathsf{pub}}$. We note that the latter task is feasible since $\mathcal{G}$ is a finite class [KLN$^+$08]. The description here is rather simplified. The actual construction and our analysis entail more intricate details (e.g., we need to carefully analyze the generalization error since the class $\mathcal{G}$ itself depends on the public part of the training set). One clear aspect from the above description is that our construction is of an improper learner since, in general, the output hypothesis is given by the intersection of at most $d$ halfspaces. Devising a construction of a proper learner for this problem is an interesting open question.

**Related work:** As mentioned earlier there have been some works that studied utilizing public data in differentially private data analysis. In particular, the notion of differentially private PAC learning assisted with public data was introduced by Beimel et al. in [BNS13], where it was called "semi-private learning." They gave a construction of a learning algorithm in this setting, and derived upper bounds on the private and public sample complexities. Alon et al. [ABM19] revisited this problem and gave nearly optimal bounds on the private and public sample complexities in the agnostic PAC model. In particular, they showed that any hypothesis class $\mathcal{H}$ can be learned using $\approx \mathsf{VC}(\mathcal{H})/\alpha^2$ private examples and $\approx \mathsf{VC}(\mathcal{H})/\alpha$ public examples where, $\mathsf{VC}(\mathcal{H})$ is the VC-dimension of $\mathcal{H}$. The work of [BCM$^+$20] introduced the related, but distinct, problem of differentially private query release assisted by public data, and gave upper and lower bounds on private and public sample complexities. In another line of work, public data was utilized to improve differentially private learning via the knowledge transfer technique [PAE$^+$17, PSM$^+$18, BTT18]. All these prior works assumed that the private and public examples arise from the same distribution (i.e., $\mathcal{D}_{\mathsf{priv}} = \mathcal{D}_{\mathsf{pub}}$), and their constructions particularly exploited this assumption. To the best of our knowledge, our work is the first to consider a formal model for learning from mixtures of private and public populations that does not entail this assumption, and our construction for halfspaces demonstrates that this assumption can be circumvented. It is also worth pointing out that, unlike the aforementioned prior work, a PPM learner is only assumed to have access to examples from the mixture distribution $\mathcal{D}$ rather than access to examples from *each* of $\mathcal{D}_{\mathsf{priv}}$ and $\mathcal{D}_{\mathsf{pub}}$. Hence, unlike prior work, our construction does not require certain number of examples from each sub-population; it only requires a certain *total* number of examples from the mixed population.

## 2   Preliminaries

In this section, we introduce some notation, state some basic concepts from learning theory, and describe some geometric properties we use throughout the paper.

**Notation:** For $n \in \mathbb{N}$, we use $[n]$ to denote the set $\{1, \ldots, n\}$. We use standard notation from the supervised learning literature (see, e.g. [SSBD14]). Let $\mathcal{X}$ denote an arbitrary domain (that represents the space of feature vectors). Let $\mathcal{Y} = \{0, 1\}$. A function $h : \mathcal{X} \to \mathcal{Y}$ is called a concept/hypothesis. A family of concepts (hypotheses) $\mathcal{C} \subseteq \mathcal{Y}^{\mathcal{X}}$ is called a concept/hypothesis class. A learning algorithm, receives as input i.i.d. samples generated from some arbitrary distribution $\mathcal{D}$ over $\mathcal{X} \times \mathcal{Y}$, and outputs a hypothesis $h \in \mathcal{Y}^{\mathcal{X}}$.

**Expected error:** The expected/population error of a hypothesis $h : \mathcal{X} \to \mathcal{Y}$ with respect to a distribution $\mathcal{D}$ over $\mathcal{X} \times \mathcal{Y}$ is defined by $\mathsf{err}(h; \mathcal{D}) \triangleq \mathbb{E}_{(x,y) \sim \mathcal{D}} [\mathbf{1}(h(x) \neq y)]$.

A distribution $\mathcal{D}$ is called *realizable by* $\mathcal{C}$ if there exists $h^* \in \mathcal{C}$ such that $\mathsf{err}(h^*; \mathcal{D}) = 0$. In this case, the data distribution $\mathcal{D}$ over $\mathcal{X} \times \mathcal{Y}$ is completely described by a distribution $\mathcal{D}_{\mathcal{X}}$ over $\mathcal{X}$ and a true labeling concept $h^* \in \mathcal{C}$.

**Empirical error:** The empirical error of an hypothesis $h : \mathcal{X} \to \{0, 1\}$ with respect to a labeled dataset $S = \{(x_1, y_1), \ldots, (x_n, y_n)\}$ will be denoted by $\widehat{\mathsf{err}}(h; S) \triangleq \frac{1}{n} \sum_{i=1}^{n} \mathbf{1}(h(x_i) \neq y_i)$.
The problem of minimizing the empirical error on a dataset (i.e. outputting an hypothesis in the class with minimal error) is known as Empirical Risk Minimization (ERM).

We next define the geometric concepts we use in this paper.

**Halfspaces and Hyperplanes:** For $\mathbf{w} = (w_0, w_1, \ldots, w_d) \in \mathbb{R}^{d+1}$, let $h_{\mathbf{w}}$ denote the halfspace defined as $h_{\mathbf{w}} \triangleq \{x \in \mathbb{R}^d : \sum_{i=1}^d w_i x_i \geq w_0\}$. We will overload the notation and use $h$ to denote both the halfspace and the corresponding binary hypothesis defined as the indicator function of the halfspace. In particular, whenever we write $h(x)$, we would be referring to $h$ as the binary hypothesis associated with the halfspace $h$, namely $h(x) \triangleq \mathbf{1}\,(x \in h)$. A pair of halfspaces $h_{\mathbf{w}}$ and $h_{-\mathbf{w}}$ will be loosely referred to as "opposite" halfspaces. A pair of opposite halfspaces $h_{\mathbf{w}}$ and $h_{-\mathbf{w}}$ intersect in the hyperplane $hp_{\mathbf{w}} \triangleq \{x \in \mathbb{R}^d : \sum_{i=1}^d w_i x_i = w_0\}$. A finite set $S \subset \mathbb{R}^d$ is said to *support* a halfspace $h_{\mathbf{w}}$ if $S$ is contained in the *hyperplane* $hp_{\mathbf{w}}$.

**Affine subspace:** A non-empty subset $\mathsf{Aff} \subseteq \mathbb{R}^d$ is an affine subspace, if there exists a $u \in \mathsf{Aff}$ such that $\mathsf{Aff} - u = \{x - u \mid x \in \mathsf{Aff}\}$ is a linear subspace of $\mathbb{R}^d$. Moreover, we say that $\mathsf{Aff}$ is $k$-dimensional affine subspace, $1 \leq k \leq d$, if the corresponding linear subspace $\mathsf{Aff} - u$ is $k$-dimensional.

Since differential privacy is central to this work, we conclude this section by stating its definition.

**Definition 2.1** (Differential Privacy [DKM$^+$06, DMNS06, DR14]). *Let $\epsilon, \delta > 0$. A (randomized) algorithm $M : \{\mathcal{X} \times \mathcal{Y}\}^n \to \mathcal{R}$ is $(\epsilon, \delta)$-differentially private if for all pairs of datasets $S, S' \in \{\mathcal{X} \times \mathcal{Y}\}$ that differ in exactly one entry, and every measurable $\mathcal{O} \subseteq \mathcal{R}$, we have:*

$$\Pr\left(M(S) \in \mathcal{O}\right) \leq e^{\epsilon} \cdot \Pr\left(M(S') \in \mathcal{O}\right) + \delta.$$

## 3 Model and Definitions

In this paper, we consider a model of privacy-preserving learning, where the input dataset is a mixture of private and public examples. We call such model *Private-Public Mixture (PPM)* learning. We view each example in the input dataset as a triplet comprised of a feature vector $x \in \mathcal{X}$, a target label $y \in \mathcal{Y}$, and a privacy status bit $p \in \mathcal{P} \triangleq \{\mathsf{priv}, \mathsf{pub}\}$. The privacy status is a bit that describes whether the example is private ($p = \mathsf{priv}$) and hence requires protection via differential privacy, or public ($p = \mathsf{pub}$) and hence does not entail any privacy concerns. In this paper, the privacy status is used only to distinguish between the private and public portions of the dataset. We stress that the goal is to learn how to classify the target label (and not the privacy bit).

In our formulation, the training examples are i.i.d. from a distribution $\mathcal{D}$ over $\mathcal{Z} \triangleq \mathcal{X} \times \mathcal{Y} \times \mathcal{P}$. Hence, the distribution $\mathcal{D}$ is a mixture of a public sub-population $\mathcal{D}_{\mathsf{pub}} \triangleq \mathcal{D}_{\mathcal{X} \times \mathcal{Y}|\mathsf{pub}}$ and private sub-population $\mathcal{D}_{\mathsf{priv}} \triangleq \mathcal{D}_{\mathcal{X} \times \mathcal{Y}|\mathsf{priv}}$, where $\mathcal{D}_{\mathcal{X} \times \mathcal{Y}|p}$ denotes the conditional distribution of the $(x, y) \in \mathcal{X} \times \mathcal{Y}$ given a privacy-status bit $p \in \mathcal{P}$. A sample $S \sim \mathcal{D}^n$ is a mixture of private and public examples that can be distinguished using the privacy-status bit. Hence, we can partition the dataset $S$ into: a private dataset $S_{\mathsf{priv}} \in (\mathcal{X} \times \mathcal{Y})^{n_{\mathsf{priv}}}$ and a public dataset $S_{\mathsf{pub}} \in (\mathcal{X} \times \mathcal{Y})^{n_{\mathsf{pub}}}$, where $n_{\mathsf{priv}} + n_{\mathsf{pub}} = n$. We note that $S_{\mathsf{priv}} \sim \mathcal{D}_{\mathsf{priv}}^{n_{\mathsf{priv}}}$ and $S_{\mathsf{pub}} \sim \mathcal{D}_{\mathsf{pub}}^{n_{\mathsf{pub}}}$.

**The PPM Learning Model:** A PPM learning model is described by the following components: (i) a distribution $\mathcal{D}$ over $\mathcal{X} \times \mathcal{Y} \times \mathcal{P}$; (ii) a dataset of $n$ i.i.d. examples from $\mathcal{D}$; (iii) a loss function $\ell : \mathcal{Y} \times \mathcal{Y} \to \mathbb{R}_+$, which we fix to be the binary loss function, i.e., $\ell(\hat{y}, y) \triangleq \mathbf{1}(\hat{y} \neq y)$, $\hat{y}, y \in \mathcal{Y}$; and (iv) a PPM learning algorithm, which we define below:

**Definition 3.1** (($\epsilon, \delta, n$)-PPM Learning Algorithm). *Let $\epsilon, \delta \in (0, 1)$, $n \in \mathbb{N}$. An $(\epsilon, \delta, n)$-PPM learning algorithm is a randomized map $\mathcal{A} : (\mathcal{X} \times \mathcal{Y} \times \mathcal{P})^n \to \mathcal{Y}^{\mathcal{X}}$ that maps datasets of size $n$ (of private and public examples) to binary hypotheses such that for any $n_{\mathsf{pub}} \leq n$ and any realization of the public portion of the input dataset $S_{\mathsf{pub}} \in (\mathcal{X} \times \mathcal{Y})^{n_{\mathsf{pub}}}$, the induced algorithm $\mathcal{A}(\cdot, S_{\mathsf{pub}})$ is $(\epsilon, \delta)$-differentially private (w.r.t. the private portion of the input dataset).*

**Expected error of a PMM algorithm $\mathcal{A}$:** Let $\tilde{\mathcal{D}}$ be a distribution over $\mathcal{X} \times \mathcal{Y}$. Let $\hat{h}$ denote the hypothesis produced by $\mathcal{A}$ on input sample $S$ of size $n$. The expected error of a PPM algorithm w.r.t. $\tilde{\mathcal{D}}$ is defined as $\mathrm{err}(\mathcal{A}(S); \tilde{\mathcal{D}}) = \mathop{\mathbb{E}}_{(x,y) \sim \tilde{\mathcal{D}}} \left[ \mathbf{1}(\hat{h}(x) \neq y) \right]$. Note that the distribution here is only over $\mathcal{X} \times \mathcal{Y}$ since, as mentioned earlier, the goal is to learn how to classify the target label and not

the privacy status. Namely, $\tilde{\mathcal{D}}$ is the distribution that is obtained from the original distribution $\mathcal{D}$ by marginalizing over the privacy status bit $p$.

Generally speaking, the goal in PPM learning is to design a PPM algorithm whose expected error is as small as possible (with high probability over the input i.i.d. sample and the algorithm's internal randomness).

As stated, the above model does not specify how we quantify the learning goal over the choice of the distribution and the sample size. This is done to maintain flexibility in defining the learning paradigm based on the PPM model. Indeed, one can make different choices about such quantifiers and their order, which would result in different modes of learnability. One standard paradigm, which we will adopt in Section 4, is to assume that the learning algorithm has access to a fixed hypothesis class $\mathcal{C} \subseteq \mathcal{Y}^{\mathcal{X}}$ and require that the algorithm attains small excess error, i.e., require that the expected error incurred by the algorithm is close to the smallest expected error attained by a hypothesis in $\mathcal{C}$ (as in (agnostic) PAC learning). However, we still need to specify how we will quantify this desired goal over the distribution $\mathcal{D}$. One possibility is to insist on uniform learnability; namely, require that we design a PPM algorithm that given a sufficiently large sample is guaranteed to have a small excess error (not exceeding a prespecified level) w.r.t. *all* distributions $\mathcal{D}$ over $\mathcal{X} \times \mathcal{Y} \times \mathcal{P}$. However, this route will lead us back to the conventional DP learning since the family of all distributions $\mathcal{D}$ clearly subsumes those distributions where all examples are private. We thus propose a meaningful alternative, where we fix a specific *conditional distribution* $\mathcal{D}_{\mathcal{P}|\mathcal{X}\times\mathcal{Y}}$ of the privacy status bit $p \in \mathcal{P}$ given labeled example $(x,y) \in \mathcal{X} \times \mathcal{Y}$ and quantify over *all* distributions $\tilde{\mathcal{D}}$ over $\mathcal{X} \times \mathcal{Y}$.

**Privacy-data model** $\mathcal{D}_{\mathcal{P}|\mathcal{X}\times\mathcal{Y}}$**:** The conditional distribution $\mathcal{D}_{\mathcal{P}|\mathcal{X}\times\mathcal{Y}}$ is given by $\left\{ \mathbb{P}\left[p = b \mid (x,y)\right] : b \in \mathcal{P}, (x,y) \in \mathcal{X} \times \mathcal{Y} \right\}$. In other words, it can be seen as a mapping taking an example $(x,y)$ to the conditional distribution of its privacy bit, $\mathbb{P}\left[p = \cdot | x, y\right]$. We refer to $\mathcal{D}_{\mathcal{P}|\mathcal{X}\times\mathcal{Y}}$ as *the privacy-data model*. Such conditional distribution captures how likely a labeled example $(x,y)$ to be sensitive (from a privacy perspective). As discussed earlier, in many practical scenarios, the attribute of being sensitive can strongly depend on the realization of the data record.

**Label-determined privacy-data model:** A special case of the above definition is when the privacy status is perfectly correlated with the target label (as in the examples discussed in the introduction and the abstract). Namely, in this case, we have $p = \mathsf{priv} \iff y = 1$ with probability 1 (or, $p = \mathsf{pub} \iff y = 1$ with probability 1). We refer to this privacy-data model as *label-determined*.

Next, we formally define one possible class of PPM learners based on the discussion above.

**Definition 3.2** (($\alpha,\beta,\epsilon,\delta$)-PPM learner for a class $\mathcal{C}$ w.r.t. a privacy-data model $\mathcal{D}_{\mathcal{P}|\mathcal{X}\times\mathcal{Y}}$)**.** *Let $\mathcal{C} \subseteq \mathcal{Y}^{\mathcal{X}}$ be a concept class, let $\mathcal{D}_{\mathcal{P}|\mathcal{X}\times\mathcal{Y}}$ be a privacy-data model, and let $\alpha, \beta, \epsilon, \delta \in (0,1)$. A randomized algorithm $\mathcal{A}$ is an ($\alpha,\beta,\epsilon,\delta$)-PPM learner for $\mathcal{C}$ w.r.t. $\mathcal{D}_{\mathcal{P}|\mathcal{X}\times\mathcal{Y}}$ with sample size $n$ if the following conditions hold:*

1. *$\mathcal{A}$ is an ($\epsilon,\delta,n$)-PPM learning algorithm (see Definition 3.1).*

2. *For every distribution $\tilde{\mathcal{D}}$ over $\mathcal{X} \times \mathcal{Y}$, given a dataset $S \sim \mathcal{D}^n$ where $\mathcal{D} = \tilde{\mathcal{D}} \times \mathcal{D}_{\mathcal{P}|\mathcal{X}\times\mathcal{Y}}$, $\mathcal{A}$ outputs a hypothesis $\hat{h}$ such that, with probability at least $1 - \beta$ (over $S \sim \mathcal{D}^n$ and the internal randomness of $\mathcal{A}$),*

$$\mathsf{err}\left(\hat{h}; \tilde{\mathcal{D}}\right) \leq \min_{h \in \mathcal{C}} \mathsf{err}\left(h; \tilde{\mathcal{D}}\right) + \alpha.$$

*When the first condition is satisfied with $\delta = 0$ (i.e., pure differential privacy), we refer to $\mathcal{A}$ as ($\alpha,\beta,\epsilon$)-PPM learner for $\mathcal{C}$ w.r.t. $\mathcal{D}_{\mathcal{P}|\mathcal{X}\times\mathcal{Y}}$.*

In the special case of *label-determined* privacy-data model, we say that $\mathcal{A}$ is an ($\alpha,\beta,\epsilon,\delta$)-PPM learner for a class $\mathcal{C}$ assuming label-determined privacy-data model.

## 4 Learning Halfspaces

We consider the problem of PPM learning for one of the most well-studied tasks in machine learning, namely, learning halfspaces (linear classifiers) in $\mathbb{R}^d$. We focus on the case of *label-determined*

*privacy-data model* defined earlier; that is, we consider the case where the privacy-status bit is perfectly correlated with the target label. In particular, a 1-labeled data point is considered to be a private data point and a 0-labeled data point is a public data point. We give a construction for a PPM learner for halfspaces in this case in both the agnostic and realizable settings. Our construction outputs a hypothesis with excess true error $\alpha$ using an input sample of size $\tilde{O}\left(\frac{d^2}{\epsilon\alpha}\right)$ in the realizable setting, and a sample of size $\tilde{O}\left(d^2 \max\left(\frac{1}{\alpha^2}, \frac{1}{\epsilon\alpha}\right)\right)$ in the agnostic setting. Our algorithm is an improper learner; specifically, the output hypothesis is given by the intersection of at most $d$ halfspaces.

**Relaxations to the label-determined privacy-data model:** Since perfect correlation between the privacy status and the target label might be a strong assumption to make in some practical scenarios, it is important for us to point out that such strict correlation is not necessary. In particular, our results (with exactly the same construction) still hold under either one of the following two relaxations to this assumption: (i) when the privacy status is only sufficiently correlated with the output label (in this case, the same construction will yield essentially the same accuracy since the impact of this relaxation on the excess error will be small); or (ii) when only the private examples have the same target label while the set of public examples can have both labels (in fact, our analysis will be exactly the same in this case). However, to emphasize the conceptual basis of our construction and maintain clarity and simplicity of the analysis, we opt to present the results for the simpler model that assumes perfect correlation.

**Overview:** The input to our private algorithm is a dataset $S \in (\mathcal{X} \times \mathcal{Y} \times \mathcal{P})^n$. The dataset $S$ is partitioned into: $S_{\mathsf{priv}} \in (\mathcal{X} \times \mathcal{Y})^{n_{\mathsf{priv}}}$ (private dataset) and $S_{\mathsf{pub}} \in (\mathcal{X} \times \mathcal{Y})^{n_{\mathsf{pub}}}$ (public dataset) using the privacy-status bit in $\mathcal{P}$, as described in Section 3, where $n_{\mathsf{priv}} + n_{\mathsf{pub}} = n$. The main idea of our algorithm is to construct a family of halfspaces in $\mathbb{R}^d$, denoted by $\widetilde{\mathcal{C}}_{\mathsf{pub}}$, using the (unlabeled) public data points, and then restrict the algorithm to a finite hypothesis class $\mathcal{G}$ made up of all intersections of at most $d$ halfspaces from $\widetilde{\mathcal{C}}_{\mathsf{pub}}$. That is, each hypothesis in the finite hypothesis class $\mathcal{G}$ is represented by an intersection of at most $d$ halfspaces from the family $\widetilde{\mathcal{C}}_{\mathsf{pub}}$. Using Helly's Theorem [Hel23, Rad21], we can show that $\mathcal{G}$ will contain one hypothesis whose error is comparable to that of the ERM halfspace. Hence, given the finite hypothesis class $\mathcal{G}$, we construct a private learner that outputs a hypothesis from $\mathcal{G}$ via the exponential mechamishm [MT07]. Our construction is described formally in Algorithm 2.

First, let's start by describing the construction of $\widetilde{\mathcal{C}}_{\mathsf{pub}}$ and the finite hypothesis class $\mathcal{G}$.

Let $\tilde{S}_{\mathsf{pub}} \in \mathcal{X}^{n_{\mathsf{pub}}}$ denote the *unlabeled* version of the public portion $S_{\mathsf{pub}}$ of the input dataset. The family of halfspaces $\widetilde{\mathcal{C}}_{\mathsf{pub}}$ is constructed as follows. Let $\mathcal{W} \triangleq \{\widehat{S} \subseteq \tilde{S}_{\mathsf{pub}} : |\widehat{S}| \leq d\}$. Namely, $\mathcal{W}$ is a collection of all the subsets of $\tilde{S}_{\mathsf{pub}}$ of at most $d$ points. Note that the size of such collection is $|\mathcal{W}| = O(n_{\mathsf{pub}}^d)$. For each $\widehat{S} \in \mathcal{W}$, we find *one* arbitrary halfspace in $\mathbb{R}^d$ that is *supported* by $\widehat{S}$, and its corresponding opposite halfspace. We add these two halfspaces to $\widetilde{\mathcal{C}}_{\mathsf{pub}}$. In addition to $\widetilde{\mathcal{C}}_{\mathsf{pub}}$, we also define the affine subspace Aff that is spanned by the points in $\tilde{S}_{\mathsf{pub}}$ (where the notion of an affine subspace is as defined in Section 2). Note that, when the points of $\tilde{S}_{\mathsf{pub}}$ are in general position, Aff is trivially taken to be the entire $\mathbb{R}^d$. The set Aff is merely needed when the public data points lie in a lower dimensional affine subspace since in this case, we can simply restrict ourselves to the intersections of the halfspaces in $\widetilde{\mathcal{C}}_{\mathsf{pub}}$ with Aff. Finally, we get a family of halfspaces $\widetilde{\mathcal{C}}_{\mathsf{pub}}$ whose size is $|\widetilde{\mathcal{C}}_{\mathsf{pub}}| = 2|\mathcal{W}| = O(n_{\mathsf{pub}}^d)$, and one additional set Aff. We remark that if there are no public examples in the dataset, (i.e., $\tilde{S}_{\mathsf{pub}} = \emptyset$), then we simply return the empty set, i.e., $\widetilde{\mathcal{C}}_{\mathsf{pub}} = \emptyset$. We formally describe the construction of $\widetilde{\mathcal{C}}_{\mathsf{pub}}$ and Aff in Algorithm 1 (denoted by $\mathcal{A}_{\mathsf{ConstrHalf}}$).

**Effective hypothesis class:** In our main algorithm $\mathcal{A}_{\mathsf{LearnHalf}}$ (Algorithm 2 below), we construct a finite hypothesis class $\mathcal{G}$ using $\widetilde{\mathcal{C}}_{\mathsf{pub}}$ described above. Each hypothesis in $\mathcal{G}$ corresponds to the intersection of at most $d$ halfspaces in the collection $\widetilde{\mathcal{C}}_{\mathsf{pub}}$ and the affine subspace Aff. Hence, it follows that $|\mathcal{G}| \leq \binom{|\widetilde{\mathcal{C}}_{\mathsf{pub}}|}{\leq d} = O(|\widetilde{\mathcal{C}}_{\mathsf{pub}}|^d) = O(2^d\, n_{\mathsf{pub}}^{d^2})$. Note that we consider the intersection of *at most* $d$ halfspaces, so $\mathcal{G}$ is assumed to also contain a hypothesis that corresponds to the empty set $\emptyset$, which assigns label 1 to all points in $\mathbb{R}^d$ (according to our definition in Step 8 of Algorithm 2).

---

**Algorithm 1** $\mathcal{A}_{\text{ConstrHalf}}$: Construction of the family $\widetilde{\mathcal{C}}_{\text{pub}}$ halfspaces

---

**Input:** Dataset: $S_{\text{pub}} \in (\mathbb{R}^d \times \mathcal{Y})^{n_{\text{pub}}}$
  1: Let $\tilde{S}_{\text{pub}}$ be the unlabeled version of $S_{\text{pub}}$.
  2: Initialize $\widetilde{\mathcal{C}}_{\text{pub}} = \emptyset$.
  3: Let $\mathcal{W} = \{\widehat{S} \subseteq \tilde{S}_{\text{pub}} : |\widehat{S}| \leq d\}$.
  4: **for** every $\widehat{S} \in \mathcal{W}$: **do**
  5:   Find a halfspace $h \in \mathbb{R}^d$ that is supported by $\widehat{S}$, and its corresponding opposite halfspace $h_-$.
       {The notion of opposite halfspaces is defined in Section 2.}
  6:   Add $h, h_-$ to $\widetilde{\mathcal{C}}_{\text{pub}}$.
  7: Let Aff be the affine subspace spanned by $\tilde{S}_{\text{pub}}$.
  8: Output $\{\widetilde{\mathcal{C}}_{\text{pub}}, \text{Aff}\}$.

---

**Algorithm 2** $\mathcal{A}_{\text{LearnHalf}}$: PPM Learning of Halfspaces

---

**Input:** Class of halfspaces in $\mathbb{R}^d$: $\mathcal{C}$; Labeled dataset: $S = \{(x_1, y_1, p_1), \ldots, (x_n, y_n, p_n)\} \in (\mathbb{R}^d \times \mathcal{Y} \times \mathcal{P})^n$, Privacy parameter: $\epsilon$
  1: Initialize $S_{\text{pub}} \leftarrow \emptyset$, $S' \leftarrow \emptyset$, $\mathcal{G} \leftarrow \emptyset$
  2: **for** $i = 1, \ldots, n$ **do**
  3:   **if** $p_i = \text{pub}$ **then**
  4:     Add $(x_i, y_i)$ to $S_{\text{pub}}$.
  5: $\{\widetilde{\mathcal{C}}_{\text{pub}}, \text{Aff}\} \leftarrow \mathcal{A}_{\text{ConstrHalf}}(S_{\text{pub}})$.
  6: **for** $i = 1, \ldots, n$ **do**
  7:   Add $(x_i, y_i)$ to $S'$     {$S'$ consists of all the $(x, y)$ pairs of $S$}
  8: For every $j \in [d]$, and every collection of distinct halfspaces $h_1, \ldots, h_j \in \widetilde{\mathcal{C}}_{\text{pub}}$, add a hypothesis $g$ to $\mathcal{G}$, where $g$ is defined as:

$$g(x) \triangleq \mathbf{1}\left(x \notin \left(\bigcap_{i=1}^{j} h_i \cap \text{Aff}\right)\right), \quad x \in \mathbb{R}^d.$$

  9: Use the exponential mechanism with inputs $S'$, $\mathcal{G}$, privacy parameter $\epsilon$, and a score function $q(S', g) \triangleq -\widehat{\text{err}}(g; S')$ to select a hypothesis $\widehat{g}$ from $\mathcal{G}$.
 10: Output $\widehat{g}$.

---

**Lemma 4.1** (Privacy Guarantee of $\mathcal{A}_{\text{LearnHalf}}$). *For any realization of the privacy-status bits $(p_1, \ldots, p_n) \in \mathcal{P}^n$, and for any realization of $S_{\text{pub}}$ constructed in Steps (2 -4) of $\mathcal{A}_{\text{LearnHalf}}$ (Algorithm 2), $\mathcal{A}_{\text{LearnHalf}}$ is $\epsilon$-differentially private (w.r.t. the private portion of the input dataset).*

The proof of the above lemma follows from the fact that $\{\widetilde{\mathcal{C}}_{\text{pub}}, \text{Aff}\}$ are constructed using only the public data together with the privacy analysis of the exponential mechanism [MT07] (see details in the full version [BMN20]).

Next, we turn to the analysis of the (excess) error of $\mathcal{A}_{\text{LearnHalf}}$. Let $h_{S'}^{\text{ERM}}$ denote the ERM halfspace with respect to the dataset $S'$; that is, $h_{S'}^{\text{ERM}} = \arg\min_{h \in \mathcal{C}} \widehat{\text{err}}(h; S')$. We will first show that the expected error of the output hypothesis of $\mathcal{A}_{\text{LearnHalf}}$ is close to that of $h_{S'}^{\text{ERM}}$. Then, we derive explicit sample complexity bounds for $\mathcal{A}_{\text{LearnHalf}}$ in the realizable and agnostic settings.

The first main step in our analysis is to show the existence of a hypothesis $g^* \in \mathcal{G}$ whose empirical error is not larger than the empirical error of $h_{S'}^{\text{ERM}}$. Let $\tilde{S}_{\text{pub}} \setminus h_{S'}^{\text{ERM}} \triangleq \{x \in \tilde{S}_{\text{pub}} : x \notin h_{S'}^{\text{ERM}}\}$. First, we consider the corner case where $\tilde{S}_{\text{pub}} \setminus h_{S'}^{\text{ERM}} = \emptyset$. In this case, all public examples are incorrectly labeled (i.e., assigned label 1) by $h_{S'}^{\text{ERM}}$. Thus, the hypothesis $g^* \in \mathcal{G}$ we are looking for is simply the empty hypothesis, which assigns label 1 to all points in $\mathbb{R}^d$. Indeed, in such case the empirical error of $g^*$ cannot be larger than that of $h_{S'}^{\text{ERM}}$ since $g^*$ correctly labels all the private examples and is consistent with $h_{S'}^{\text{ERM}}$ on all the public examples.

Thus, in the remainder of our analysis, we will assume w.l.o.g. that $\tilde{S}_{\mathsf{pub}} \setminus h_{S'}^{\mathsf{ERM}} \neq \emptyset$. We first state some useful facts from convex geometry.

For any finite set $T \subset \mathbb{R}^d$, we use $\mathcal{V}(T)$ to denote the convex hull of all the data points in $T$. Note that $\mathcal{V}(T)$ is a convex polytope that is given by the intersection of at most $O(|T|^d)$ halfspaces. Hence, $\mathcal{V}(\tilde{S}_{\mathsf{pub}} \setminus h_{S'}^{\mathsf{ERM}})$ is a convex polytope that contains all the public data points that are labeled correctly by $h_{S'}^{\mathsf{ERM}}$. Moreover, $\mathcal{V}(\tilde{S}_{\mathsf{pub}} \setminus h_{S'}^{\mathsf{ERM}})$ is given by the intersection of a sub-collection of halfspaces in $\widetilde{\mathcal{C}}_{\mathsf{pub}}$ and Aff. (As mentioned earlier, intersection with Aff is needed only when all the public data points lie in a lower dimensional affine subspace. In this case, the convex hull $\mathcal{V}(\tilde{S}_{\mathsf{pub}} \setminus h_{S'}^{\mathsf{ERM}})$ is a "flat" set that lies in this affine subspace.) Thus, we can make the following immediate observation:

**Observation 4.2.** *Let $h_1, \ldots, h_v$ be halfspaces in $\widetilde{\mathcal{C}}_{\mathsf{pub}}$ such that $\left( \bigcap_{i=1}^{v} h_i \right) \cap \mathsf{Aff} = \mathcal{V}(\tilde{S}_{\mathsf{pub}} \setminus h_{S'}^{\mathsf{ERM}})$. Then $\left( \left( \bigcap_{i=1}^{v} h_i \right) \cap \mathsf{Aff} \right) \cap h_{S'}^{\mathsf{ERM}} = \emptyset$.*

A key step in our analysis relies on an application of a basic result in convex geometry known as Helly's Theorem, which we state below.

**Lemma 4.3** (Helly's Theorem restated [Hel23, Rad21]). *Let $N \in \mathbb{N}$. Let $\mathcal{F} = \{C_1, C_2, \ldots, C_N\}$ be a family of convex sets in $\mathbb{R}^d$. Suppose we have $\bigcap_{i=1}^{N} C_i = \emptyset$, then there exists a collection $C_{i_1}, \ldots, C_{i_K}$, where $K \leq d+1$, such that $C_{i_1} \cap \ldots \cap C_{i_K} = \emptyset$.*

Combining Observation 4.2 and Lemma 4.3, we obtain the following corollary:

**Corollary 4.4.** *There exists a sub-collection of sets $\mathcal{T} \subseteq \widetilde{\mathcal{C}}_{\mathsf{pub}} \cup \{\mathsf{Aff}\}$, where $|\mathcal{T}| \leq d$, such that $\left( \bigcap_{h \in \mathcal{T}} h \right) \cap h_{S'}^{\mathsf{ERM}} = \emptyset$.*

*Proof.* By Lemma 4.3 and Observation 4.2, there exists a sub-collection $\mathcal{T}' \subseteq \{h_1, \ldots, h_v, \mathsf{Aff}, h_{S'}^{\mathsf{ERM}}\}$ of size $|\mathcal{T}'| \leq d+1$ such that the intersection of the sets in $\mathcal{T}'$ is empty (where $h_1, \ldots, h_v$ are the halfspaces in Observation 4.2). Observe that necessarily $h_{S'}^{\mathsf{ERM}} \in \mathcal{T}'$ since $\left( \bigcap_{i=1}^{v} h_i \right) \cap \mathsf{Aff} = \mathcal{V}(\tilde{S}_{\mathsf{pub}} \setminus h_{S'}^{\mathsf{ERM}}) \neq \emptyset$. Therefore $\mathcal{T} = \mathcal{T}' \setminus \{h_{S'}^{\mathsf{ERM}}\}$ gives the desired collection. $\qquad\square$

Define $g^*(x) \triangleq \mathbf{1}\left( x \notin \bigcap_{h \in \mathcal{T}} h \right)$, $x \in \mathbb{R}^d$, where $\mathcal{T}$ is the collection of at most $d$ sets whose existence is established in Corollary 4.4. Note that $g^* \in \mathcal{G}$. Given this definition of $g^*$, we note that all points in $S'$ that are labeled correctly by $h_{S'}^{\mathsf{ERM}}$ are also labeled correctly by $g^*$. Indeed, for any private (i.e. 1-labeled) data point $x$ that $h_{S'}^{\mathsf{ERM}}$ labels correctly (i.e. $x \in h_{S'}^{\mathsf{ERM}}$), we have $x \notin \left( \bigcap_{h \in \mathcal{T}} h \right)$ by Corollary 4.4. Hence, $g^*$ labels $x$ correctly. Conversely, for any public (i.e., 0-labeled) data point $x$ that $h_{S'}^{\mathsf{ERM}}$ labels correctly (i.e. $x \notin h_{S'}^{\mathsf{ERM}}$), we must have $x \in \mathcal{V}(\tilde{S}_{\mathsf{pub}} \setminus h_{S'}^{\mathsf{ERM}}) \subseteq \left( \bigcap_{h \in \mathcal{T}} h \right)$, where the last step follows from the definition of the collection $\mathcal{T}$ in the proof of Corollary 4.4. Hence, $g^*$ also labels $x$ correctly. This clearly implies that $\widehat{\mathsf{err}}(g^*; S') \leq \widehat{\mathsf{err}}(h_{S'}^{\mathsf{ERM}}; S')$.

Next, using the fact above together with the standard accuracy analysis of the exponential mechanism [MT07, KLN$^+$08], in the following claim we show that, with high probability, the empirical error of output hypothesis $\widehat{g}$ of $\mathcal{A}_{\mathsf{LearnHalf}}$ is close to that of $h_{S'}^{\mathsf{ERM}}$. The full details are deferred to the full version [BMN20].

**Claim 4.5** (Excess Empirical Error of $\mathcal{A}_{\mathsf{LearnHalf}}$). *Let $\alpha, \beta, \epsilon \in (0,1)$. Let $S' \in (\mathbb{R}^d \times \mathcal{Y})^n$ be any realization of the dataset. For $n = O\left( \frac{d^2 \log(d/\epsilon\alpha) + \log(1/\beta)}{\epsilon \, \alpha} \right)$, with probability at least $1 - \beta$ (over the randomness Step 9 of $\mathcal{A}_{\mathsf{LearnHalf}}$), $\mathcal{A}_{\mathsf{LearnHalf}}$ outputs a hypothesis $\widehat{g} \in \mathcal{G}$ that satisfies:*

$$\widehat{\mathsf{err}}(\widehat{g}; S') - \widehat{\mathsf{err}}(h_{S'}^{\mathsf{ERM}}; S') \leq \alpha.$$

Now the remaining ingredient in our analysis is to show that the generalization error of $\mathcal{A}_{\mathsf{LearnHalf}}$ is also small. We do this by observing that each hypothesis in $\mathcal{G}$ can be described by a few data points from the input dataset and then invoking standard sample compression bounds [LW86, SSBD14]. Specifically, we observe that each $g \in \mathcal{G}$ is an intersection of at most $d$ halfspaces in $\widetilde{\mathcal{C}}_{\mathsf{pub}}$ (restricted to Aff), and each one of these halfspaces is represented by at most $d$ points from the input dataset. Hence, by putting all the ingredients together, we can finally arrive at our main result formally stated in the theorem below. Due to space considerations, we defer the details of the sample compression argument and the full proof of the theorem below to the full version [BMN20].

**Theorem 4.6** (PPM learning of halfspaces). *Let $\epsilon, \alpha, \beta \in (0, 1)$. Assuming label-determined privacy-data model, $\mathcal{A}_{\mathsf{LearnHalf}}$ (Algorithm 2) is an $(\alpha, \beta, \epsilon)$-PPM learner for halfspaces in $\mathbb{R}^d$, with input sample size*

$$n = O\left(\left(d^2 \log\left(\frac{d}{\epsilon\alpha}\right) + \log(\frac{1}{\beta})\right) \max\left(\frac{1}{\alpha^2}, \frac{1}{\epsilon\,\alpha}\right)\right).$$

*Moreover, if we assume realizability, then $\mathcal{A}_{\mathsf{LearnHalf}}$ is an $(\alpha, \beta, \epsilon)$-PPM learner for halfspaces in $\mathbb{R}^d$ with input sample size:*

$$n = O\left(\frac{d^2 \log(d/\epsilon\alpha) + \log(1/\beta)}{\epsilon\,\alpha}\right).$$

## 5   Discussion

The main goal of this work is to introduce a new, more flexible framework for differentially private learning that captures more realistic scenarios than prior works. Although the label-determined privacy-data model that we assume for our results on learning halfspaces may seem a bit restrictive, we want to point out that this model in fact captures some realistic scenarios. For example, imagine a scenario where the data of individuals who tested positive for COVID-19 does not require privacy protection (to enable contact tracing and symptom analysis), while the data of those who tested negative remains protected. This scenario is exactly captured by the label-determined model.

There are various future directions that one may explore based on our general framework. Perhaps, among the most realistic ones are those based on a distribution-dependent model, where one also restricts the data distribution (and not only the concept class), or privacy models where the privacy status can be correlated with the feature vector not just the target label. Also, in the distribution-independent setting one may ask whether every VC class can be learned in the label-determined setting. We note that in the other extreme, where the privacy status and the label are independent, previous works [BNS13, ABM19] showed that every VC class is learnable, with significant savings in sample complexity. Moreover, one may also consider privacy models that interpolate between these two extremes (label-determined and label-independent), and explore the sample complexity in this spectrum.

### Broader Impact

Our work is theoretical in nature. Although there are no concrete, foreseeable ethical or societal impact for the research presented here, we hope that the framework we present for learning from mixtures of private and public populations could provide new insights that lead to a more realistic modeling for the problem of learning under privacy constraints. In particular, we believe that our framework can be a basis for a more general framework that captures and exploits the heterogeneous nature of privacy constraints across a population. This, in turn, can lead to new practical privacy-preserving learning algorithms that meaningfully exploit data with no (or weak) privacy concerns while providing strong privacy protection for data of more sensitive nature. Making progress in this direction can have significant impact on society in the long term.

### Acknowledgments and Disclosure of Funding

RB's and AN's research is supported by NSF Award AF-1908281, Google Faculty Research Award, and OSU faculty start-up support. SM's research is supported by the Israel Science Foundation (grant No. 1225/20), by an Azrieli Faculty Fellowship, and by a grant from the United States - Israel Binational Science Foundation (BSF).

## Footnotes

[1][BNSV15, ALMM18] showed that the class of one-dimensional halfspaces over any finite domain $X \subseteq \mathbb{R}$ requires sample complexity at least $\Omega\left(\log^{\star} |X|\right)$.

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
