[Supplementary Material]

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

1: Let $\tilde{S}_{\mathsf{pub}}$ be the unlabeled version of $S_{\mathsf{pub}}$.
2: Initialize $\widetilde{\mathcal{C}}_{\mathsf{pub}} = \emptyset$.
3: Let $\mathcal{W} = \{\widehat{S} \subseteq \tilde{S}_{\mathsf{pub}} : |\widehat{S}| \leq d\}$.
4: **for** every $\widehat{S} \in \mathcal{W}$: **do**
5:     Find a halfspace $h \in \mathbb{R}^d$ that is supported by $\widehat{S}$, and its corresponding opposite halfspace $h_-$. {The notion of opposite halfspaces is defined in Section 2.}
6:     Add $h, h_-$ to $\widetilde{\mathcal{C}}_{\mathsf{pub}}$.
7: Let Aff be the affine subspace spanned by $\tilde{S}_{\mathsf{pub}}$.
8: Output $\{\widetilde{\mathcal{C}}_{\mathsf{pub}}, \text{Aff}\}$.

---

**Effective hypothesis class:** In our main algorithm $\mathcal{A}_{\mathsf{LearnHalf}}$ (Algorithm 2 below), we construct a finite hypothesis class $\mathcal{G}$ using $\widetilde{\mathcal{C}}_{\mathsf{pub}}$ described above. Each hypothesis in $\mathcal{G}$ corresponds to the intersection of at most $d$ halfspaces in the collection $\widetilde{\mathcal{C}}_{\mathsf{pub}}$ and the affine subspace Aff. Hence, it follows that $|\mathcal{G}| \leq \binom{|\widetilde{\mathcal{C}}_{\mathsf{pub}}|}{\leq d} = O(|\widetilde{\mathcal{C}}_{\mathsf{pub}}|^d) = O(2^d\, n_{\mathsf{pub}}^{d^2})$. Note that we consider the intersection of *at most* $d$ halfspaces, so $\mathcal{G}$ is assumed to also contain a hypothesis that corresponds to the empty set $\emptyset$, which assigns label 1 to all points in $\mathbb{R}^d$ (according to our definition in Step 8 of Algorithm 2). The privacy guarantee of $\mathcal{A}_{\mathsf{LearnHalf}}$ is given by the following lemma.

---

**Algorithm 2** $\mathcal{A}_{\mathsf{LearnHalf}}$: PPM Learning of Halfspaces

---

**Input:** Class of halfspaces in $\mathbb{R}^d$: $\mathcal{C}$; Labeled dataset: $S = \{(x_1, y_1, p_1), \ldots, (x_n, y_n, p_n)\} \in (\mathbb{R}^d \times \mathcal{Y} \times \mathcal{P})^n$, Privacy parameter: $\epsilon$

1: Initialize $S_{\mathsf{pub}} \leftarrow \emptyset$, $S' \leftarrow \emptyset$, $\mathcal{G} \leftarrow \emptyset$
2: **for** $i = 1, \ldots, n$ **do**
3:    **if** $p_i = \mathsf{pub}$ **then**
4:       Add $(x_i, y_i)$ to $S_{\mathsf{pub}}$.
5: $\{\widetilde{\mathcal{C}}_{\mathsf{pub}}, \mathsf{Aff}\} \leftarrow \mathcal{A}_{\mathsf{ConstrHalf}}(S_{\mathsf{pub}})$.
6: **for** $i = 1, \ldots, n$ **do**
7:    Add $(x_i, y_i)$ to $S'$    $\{S'$ consists of all the $(x, y)$ pairs of $S\}$
8: For every $j \in [d]$, and every collection of distinct halfspaces $h_1, \ldots, h_j \in \widetilde{\mathcal{C}}_{\mathsf{pub}}$, add a hypothesis $g$ to $\mathcal{G}$, where $g$ is defined as:

$$g(x) \triangleq \mathbf{1}\left(x \notin \left(\bigcap_{i=1}^{j} h_i \cap \mathsf{Aff}\right)\right), \quad x \in \mathbb{R}^d.$$

9: Use the exponential mechanism with inputs $S'$, $\mathcal{G}$, privacy parameter $\epsilon$, and a score function $q(S', g) \triangleq -\widehat{\mathsf{err}}(g; S')$ to select a hypothesis $\widehat{g}$ from $\mathcal{G}$.
10: Output $\widehat{g}$.

---

**Lemma 4.1** (Privacy Guarantee of $\mathcal{A}_{\mathsf{LearnHalf}}$). *For any realization of the privacy-status bits $(p_1, \ldots, p_n) \in \mathcal{P}^n$, and for any realization of $S_{\mathsf{pub}}$ constructed in Steps (2 -4) of $\mathcal{A}_{\mathsf{LearnHalf}}$ (Algorithm 2), $\mathcal{A}_{\mathsf{LearnHalf}}$ is $\epsilon$-differentially private (w.r.t. the private portion of the input dataset).*

*Proof.* For any $S_{\mathsf{pub}} \in (\mathcal{X} \times \mathcal{Y})^{n_{\mathsf{pub}}}$, the family of halfspaces $\widetilde{\mathcal{C}}_{\mathsf{pub}}$ and the affine subspace $\mathsf{Aff}$ (Step 5 in Algorithm 2) are constructed using only the public part of the dataset $S$ (and hence, so is $\mathcal{G}$). The private part of $S$ is invoked in Step 9, which is an instantiation of the exponential mechanism. Thus, the proof follows directly from the privacy guarantee of the exponential mechanism [MT07]. $\square$

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

**Lemma 4.5.** *There exists a hypothesis $g^* \in \mathcal{G}$ that satisfies*

$$\widehat{\text{err}}(g^*; S') \leq \widehat{\text{err}}(h_{S'}^{\text{ERM}}; S').$$

Next, using the properties of the exponential mechanism, we can show that with high probability the empirical error of output hypothesis $\widehat{g}$ of $\mathcal{A}_{\text{LearnHalf}}$ is close to that of $g^*$.

**Lemma 4.6.** *Let $\alpha, \beta, \epsilon \in (0,1)$. Let $S' \in (\mathbb{R}^d \times \mathcal{Y})^n$ be any realization of the dataset. For $n = O\left( \frac{d^2 \log(d/\epsilon\alpha) + \log(1/\beta)}{\epsilon\,\alpha} \right)$, with probability at least $1 - \beta$ (over the randomness Step 9 of $\mathcal{A}_{\text{LearnHalf}}$), $\mathcal{A}_{\text{LearnHalf}}$ outputs a hypothesis $\widehat{g} \in \mathcal{G}$ that satisfies:*

$$\widehat{\text{err}}(\widehat{g}; S') - \widehat{\text{err}}(g^*; S') \leq \alpha.$$

*Proof.* Note that $|\mathcal{G}| = O(2^d \, n_{\text{pub}}^{d^2})$, and that the score function for the exponential mechanism is $-\widehat{\text{err}}(h; S')$, whose global sensitivity is $1/n$.

By standard accuracy guarantees of exponential mechanism [MT07], it follows that an input sample size

$$n = O\left( \frac{1}{\epsilon\alpha} \left( \log\left(|\mathcal{G}|\right) + \log\left(\frac{1}{\beta}\right) \right) \right)$$

is sufficient to ensure that, w.p. $\geq 1 - \beta$ (over the randomness Step 9), we have

$$\widehat{\text{err}}(\widehat{g}; S') \leq \min_{g \in \mathcal{G}} \widehat{\text{err}}(g; S') + \alpha,$$

which implies that $\widehat{\text{err}}(\widehat{g}; S') \leq \widehat{\text{err}}(g^*; S') + \alpha$.

Substituting the size of $\mathcal{G}$, it follows that

$$n = O\left(\frac{1}{\epsilon\alpha}\left(\log\left(|\mathcal{G}|\right) + \log\left(\frac{1}{\beta}\right)\right)\right) = O\left(\frac{1}{\epsilon\alpha}\left(\log\left(2^d\, n_{\mathsf{pub}}^{d^2}\right) + \log\left(\frac{1}{\beta}\right)\right)\right)$$
$$= O\left(\frac{1}{\epsilon\alpha}\left(d^2\log\left(\frac{d}{\epsilon\alpha}\right) + \log\left(\frac{1}{\beta}\right)\right)\right).$$

$\square$

By combining the two previous lemmas, we directly reach the following claim that asserts that the empirical error of the output hypothesis of $\mathcal{A}_{\mathsf{LearnHalf}}$ is close to that of the ERM halfspace $h_{S'}^{\mathsf{ERM}} \in \mathcal{C}$.

**Claim 4.7** (Excess Empirical Error of $\mathcal{A}_{\mathsf{LearnHalf}}$). *Let $\alpha, \beta, \epsilon \in (0,1)$. Let $S' \in (\mathbb{R}^d \times \mathcal{Y})^n$ be any realization of the dataset. For $n = O\left(\frac{d^2\log(d/\epsilon\alpha) + \log(1/\beta)}{\epsilon\,\alpha}\right)$, with probability at least $1 - \beta$ (over the randomness Step 9 of $\mathcal{A}_{\mathsf{LearnHalf}}$), $\mathcal{A}_{\mathsf{LearnHalf}}$ outputs a hypothesis $\widehat{g} \in \mathcal{G}$ that satisfies:*

$$\widehat{\mathsf{err}}\left(\widehat{g}; S'\right) - \widehat{\mathsf{err}}(h_{S'}^{\mathsf{ERM}}; S') \leq \alpha.$$

Now the remaining ingredient in our analysis is to show that the generalization error of $\mathcal{A}_{\mathsf{LearnHalf}}$ is also small. In fact, we will show that this is indeed the case for any algorithm that outputs a hypothesis in $\mathcal{G}$. We observe that each hypothesis in $\mathcal{G}$ is an intersection of at most $d$ halfspaces in $\tilde{\mathcal{C}}_{\mathsf{pub}}$ (possibly restricted to a lower dimensional affine subspace), and each one of these halfspaces is represented by at most $d$ points from the input dataset (Step 5 in Algorithm 1). Hence, by using standard sample compression bounds [LW86, SSBD14], we can derive a bound on the generalization error of any algorithm that outputs any hypothesis in $\mathcal{G}$.

**Lemma 4.8** (Sample Compression bound restated [LW86, SSBD14]). *Let $k$ be an integer and let $\mathcal{B} : (\mathcal{X} \times \mathcal{Y})^k \to \mathcal{G}$ be a mapping from sequences of $k$ examples to the hypothesis class $\mathcal{G}$. Let $\mathcal{A} : (\mathcal{X} \times \mathcal{Y})^n \to \mathcal{G}$ be a learning rule that takes as input a dataset $S = ((x_1, y_1), \ldots, (x_n, y_n))$, and returns a hypothesis such that $\mathcal{A}(S) = \mathcal{B}((x_{i_1}, y_{i_1}), \ldots, (x_{i_k}, y_{i_k}))$ for some set of indices $(i_1, \ldots, i_k) \in [n]^k$. Then for any distribution $\tilde{\mathcal{D}}$ over $\mathcal{X} \times \mathcal{Y}$, with probability at least $1 - \beta$ (over $S \sim \tilde{\mathcal{D}}^n$), we have:*

$$\left|\mathsf{err}(\mathcal{A}(S); \tilde{\mathcal{D}}) - \widehat{\mathsf{err}}(\mathcal{A}(S); S)\right| \leq \sqrt{\widehat{\mathsf{err}}(\mathcal{A}(S); S)\frac{4k\log(n/\beta)}{n}} + \frac{8k\log(n/\beta)}{n} + \frac{2k}{n}.$$

Now we have all the ingredients to state and prove sample complexity bounds for our construction in both the realizable and agnostic settings. In the following statements, note that we already proved the privacy guarantee of $\mathcal{A}_{\mathsf{LearnHalf}}$ in Lemma 4.1, and so we only focus on proving the sample complexity bounds.

**Theorem 4.9** (PPM learning of halfspaces in the realizable case). *Let $\alpha, \beta, \epsilon \in (0,1)$. Assuming realizability, and assuming label-determined privacy model, $\mathcal{A}_{\mathsf{LearnHalf}}$ (Algorithm 2) is an $(\alpha, \beta, \epsilon)$-PPM learner for halfspaces in $\mathbb{R}^d$ with input sample size:*

$$n = O\left(\frac{d^2\log(d/\epsilon\alpha) + \log(1/\beta)}{\epsilon\,\alpha}\right).$$

*Proof.* Let $\tilde{\mathcal{D}}$ be any distribution over $\mathcal{X} \times \mathcal{Y}$. Suppose $S' \sim \tilde{\mathcal{D}}^n$ (where $S'$ is a dataset in Step 7 of Algorithm 2). Note that by Claim 4.7, we get that w.p. $\geq 1 - \beta/2$ (over randomness in Step 9 of Algorithm 2)

$$\widehat{\mathsf{err}}\left(\widehat{g}; S'\right) - \widehat{\mathsf{err}}(h_{S'}^{\mathsf{ERM}}; S') = \widehat{\mathsf{err}}\left(\widehat{g}; S'\right) \leq \frac{\alpha}{2}$$

as long as $n = O\left(\frac{d^2\log(d/\epsilon\alpha) + \log(1/\beta)}{\epsilon\,\alpha}\right)$, where here we used the fact that $\widehat{\mathsf{err}}(h_{S'}^{\mathsf{ERM}}; S') = 0$ since this is the realizable setting.

Note that Lemma 4.8 (together with the argument before the statement of the lemma) immediately yields a bound on the generalization error of $\mathcal{A}_{\mathsf{LearnHalf}}$ (with $k = d^2$ in the statement of the lemma). Namely, with probability $\geq 1 - \beta/2$ (over the choice of $S' \sim \tilde{\mathcal{D}}^n$ and randomness in Step 9 of Algorithm 2), we have:

$$|\mathsf{err}(\widehat{g}; \tilde{\mathcal{D}}) - \widehat{\mathsf{err}}(\widehat{g}; S')| \leq \sqrt{\widehat{\mathsf{err}}(\widehat{g}; S)\frac{4d^2 \log(2n/\beta)}{n}} + \frac{8d^2 \log(2n/\beta)}{n} + \frac{2d^2}{n}.$$

Now, using the bound on $\widehat{\mathsf{err}}(\widehat{g}; S')$ above and for $n = O\left(\frac{d^2 \log(d/\epsilon\alpha) + \log(1/\beta)}{\epsilon\,\alpha}\right)$, we conclude that w.p. $\geq 1 - \beta$, (over $S' \sim \tilde{\mathcal{D}}^n$ and the randomness in $\mathcal{A}_{\mathsf{LearnHalf}}$), we have: $\mathsf{err}\left(\widehat{g}; \tilde{\mathcal{D}}\right) \leq \alpha$. $\qquad\square$

**Theorem 4.10** (PPM learning of halfspaces in the agnostic case)**.** *Let $\epsilon, \alpha, \beta \in (0,1)$. Assuming label-determined privacy model, $\mathcal{A}_{\mathsf{LearnHalf}}$ (Algorithm 2) is an $(\alpha, \beta, \epsilon)$-PPM learner for halfspaces in $\mathbb{R}^d$, with input sample size*

$$n = O\left(\left(d^2 \log\left(\frac{d}{\epsilon\alpha}\right) + \log(\frac{1}{\beta})\right) \max\left(\frac{1}{\alpha^2}, \frac{1}{\epsilon\,\alpha}\right)\right).$$

*Proof.* As in the proof of Theorem 4.9, with probability $\geq 1 - \beta/4$ (over the randomness in Step 9 of $\mathcal{A}_{\mathsf{LearnHalf}}$), we have:

$$\widehat{\mathsf{err}}\left(\widehat{g}; S'\right) - \widehat{\mathsf{err}}(h_{S'}^{\mathsf{ERM}}; S') \leq \frac{\alpha}{4} \tag{1}$$

as long as $n = O\left(\frac{d^2 \log(d/\epsilon\alpha) + \log(1/\beta)}{\epsilon\,\alpha}\right)$. As before, Lemma 4.8 implies that with probability $\geq 1 - \beta/4$ (over the randomness in $S'$ and in $\mathcal{A}_{\mathsf{LearnHalf}}$), we have

$$|\mathsf{err}(\widehat{g}; \tilde{\mathcal{D}}) - \widehat{\mathsf{err}}(\widehat{g}; S')| \leq \frac{8d^2 \log(4n/\beta)}{n} + \frac{2d^2}{n}.$$

Hence, for $n = O\left(\frac{d^2 \log(d/\alpha) + \log(1/\beta)}{\alpha^2}\right)$, with probability $\geq 1 - \beta/4$, we have:

$$|\mathsf{err}(\widehat{g}; \tilde{\mathcal{D}}) - \widehat{\mathsf{err}}(\widehat{g}; S')| \leq \alpha/4, \tag{2}$$

Moreover, by standard uniform convergence bounds [SSBD14], for $n = O(\frac{d \log(1/\alpha) + \log(1/\beta)}{\alpha^2})$, with probability $\geq 1 - \beta/2$ (over $S' \sim \tilde{\mathcal{D}}^n$), we have:

$$\left|\mathsf{err}\left(h_{S'}^{\mathsf{ERM}}; \tilde{\mathcal{D}}\right) - \widehat{\mathsf{err}}\left(h_{S'}^{\mathsf{ERM}}; S'\right)\right| \leq \alpha/4 \tag{3}$$

$$\mathsf{err}\left(h_{S'}^{\mathsf{ERM}}; \tilde{\mathcal{D}}\right) - \min_{h \in \mathcal{C}} \mathsf{err}(h; \tilde{\mathcal{D}}) \leq \frac{\alpha}{4} \tag{4}$$

Finally by combining (1)-(4), and by the triangle inequality and the union bound, we conclude that for $n = O\left(\left(d^2 \log\left(\frac{d}{\epsilon\alpha}\right) + \log(\frac{1}{\beta})\right) \max\left(\frac{1}{\alpha^2}, \frac{1}{\epsilon\,\alpha}\right)\right)$, with probability $\geq 1 - \beta$ (over the randomness in $S'$ and $\mathcal{A}_{\mathsf{LearnHalf}}$), we have $\mathsf{err}\left(\widehat{g}; \tilde{\mathcal{D}}\right) - \min_{h \in \mathcal{C}} \mathsf{err}(h; \tilde{\mathcal{D}}) \leq \alpha$. $\qquad\square$