[Reviews · NeurIPS 2020]

Review 1

Summary and Contributions: This paper considers learning from mixtures of private and public populations -- where some individuals require privacy of their data and some do not. The data content may also be correlated with the privacy requirement, so it is not sufficient to learn only on one population, or to treat these as iid samples. For example, in a medical study, healthy individuals may not require privacy of their data since its "not embarrassing", and sick individuals may require strong differential privacy guarantees. The authors provide a formal model and definition for this new setting, and show how to learn halfspaces in this model.

Strengths: I support accepting this paper. To the best of my knowledge, it is the first work to formalize this problem. It addresses a novel technical problem of practical importance, introduces a new model and accompanying definitions, provides solid technical results, and is well written.

Weaknesses: Minor comments: The privacy model introduced on page 5 was confusing because of its name. It sounds like a new privacy definition or an attack model, when it was actually describing the conditional distribution of privacy bits.

Correctness: Yes

Clarity: Yes

Relation to Prior Work: Yes

Reproducibility: Yes

Additional Feedback: EDIT: I have read the authors' response, and I remain enthusiastic about accepting the paper. I think the authors did a great job of responding to comments from all three reviewers.


Review 2

Summary and Contributions: This paper studies the problem of supervised learning from mixtures of private the public populations when the private and public data are drawn from different subpopulations. This paper focuses on learning halfspaces, and provides an differentially private algorithm. They also give sample complexity analysis for the algorithm under the label-determined privacy model, and show that the sample complexity is comparable to that of the non-private PAC learning.

Strengths: The paper consider an interesting and practical angle that the private and public data are from different populations. I can see lots of extensions along this direction. The algorithm is clean and the sample complexity analysis is tight.

Weaknesses: I think the label-determined privacy model is too restricted. Intuitively, the theoretical sample complexity should be comparable to that of the non-private one, since the additional noise only added to the private data that behaves very differently. I'd expect some simple but a bit more advance model, like a bernoulli model characterizes private status and the target label, and the sample complexity accounts for the parameter p.

Correctness: I didn't fully examine the sample complexity but the sketch seems correct.

Clarity: This is a well-written paper.

Relation to Prior Work: Mostly yes. I'd like to see a short discussion on the line of work on private knowledge transfer, and private learning halfspaces.

Reproducibility: Yes

Additional Feedback: EDIT: The authors made good response. I am convinced that the sample complexity under the label-determined model is non-trivial. I'm happy to increase my score. One more additional comment (just for future work) is that I'm curious whether reweighting in knowledge transfer would be useful in this setting when the two distributions are different.


Review 3

Summary and Contributions: The paper studies a new model of privacy for supervised learning. The authors propose a model where the dataset is made up of samples from a mixture of a private data distribution and a public data distribution and the goal of the learning paradigm is to minimize excess error over the marginal distribution (marginalizing over the privacy status) while satisfying the differential privacy constraints only with respect to the private data and they call it Private-Public Mixture (PPM) learning. The authors also give an improper learning algorithm for learning halfspaces under the PPM learning model under the assumption that the fact that the data is private or not is perfectly correlated with the just label of the data and they call this assumption label-determined privacy model, for example the data with label 1 is private and with label 0 is not private. The authors give sample complexity results for excess risk and the hypothesis returned is a combination of up to $d$ half-spaces (because it’s an improper learner). The authors do mention that similar guarantees also hold under small relaxations to this label-determined privacy model.

Strengths: The proposed model of preserving privacy with respect to just the private data but having access to public data where no privacy concerns are required is not exactly new as prior works [eg Bassily et al’ 18] have also considered the same. The novelty in this work is that the model assumes different distributions for private and public data whereas all prior works assumed the public data was from the same distribution as the private data and I do believe this model has novelty. The authors motivate why this model of privacy requirement with different distributions for private and public data is realistic using a couple of examples about health data, and credit score data. The authors give an intuitive framework for learning under this setting which they formalize as PPM learning. The sample complexity results for half-spaces under the label-determined privacy model also has some merit as it shows the existence of a learner (albeit improper) under the PPM learning framework but I do feel the label-determined privacy model is very restrictive. ------------------------------------------------------------------------------- Edit after author feedback: I have read the rebuttal. I do agree that there is definitely novelty in the setting proposed by the authors and since it's the first result, I can let the restricted label-determined privacy model slide. I do believe this work can inspire a lot of future work.

Weaknesses: The extra novelty in the new privacy framework introduced in this work is only that the distribution of the private and public data is different as using public data to aid private learning has already been studied before. The results for the halfspaces are under the very restrictive label-determined privacy settings and I wonder what kind of results are even possible without making such assumptions about the privacy model. The learning algorithm given for halfspaces is an improper learning algorithm (i.e. the hypothesis returned may not be in the hypothesis class being compared to) and is also not polytime in the dimension as the exponential mechanism requires sampling from $O(2^d . n^{d^2})$ hypothesis and I wonder if there is an efficient algorithm for the same. The sample complexity given is also O(d^2) and I wonder if it’s possible to get something close to the non-private case of only O(d). A discussion about it would have been appreciated. The tools used for proving the half-space case are also specialized and these ideas may not generalize to general classes of bounded VC dimension.

Correctness: The claims are correct as per my understanding. There are no experiments in the paper.

Clarity: The paper is well written for a draft and was easy to follow. The setting was well motivated and the appropriate background information has been provided to understand the context and follow the paper.

Relation to Prior Work: Yes, authors do discuss prior work about learning while preserving privacy and about prior work on learning with public and private data and how those settings differ from the one studied there. Maybe comparison/discussion for the sample complexity of halfspaces with the non-private case and the private/public but with the same distribution setting should have been included.

Reproducibility: Yes

Additional Feedback:

[Author Response · NeurIPS 2020]

We would like to thank the reviewers for their comments and helpful suggestions. We respond below to the reviewers' comments and criticism.

- *R1: The term "privacy model" is confusing:* We will definitely think of a better term. Perhaps, "privacy-data" model is less confusing.

- *R1: More discussion of the broader impact"* We will elaborate more on the potential impact of the proposed framework. We believe it can be a basis for a more flexible, more expressive framework for DP learning.

- *R3: "Intuitively, the sample complexity should be comparable to that of the non-private one":* We disagree with the reviewer's intuition about the problem. The question of the sample complexity under this model is quite different from its non-private counterpart. In particular, our construction is tailored to halfspaces, and – as R4 pointed out – extending this to general VC classes is non-trivial (and is an open question).

- *R3: "I'd expect some simple but a bit more advance model, like a Bernoulli model ... ":* The general framework enables studying settings such as the one suggested by the reviewer (as well as many others). In this work, we focus on this special setting (label-determined) to demonstrate the capacity of the proposed framework to capture realistic settings, which were not explored by previous works that studied utilizing public data.

- *R3 & R4: label-determined privacy model is too restrictive:* The main goal of this work is to propose a new, formal, more flexible framework for DP learning that captures more realistic scenarios than prior works. The result concerning the label-determined model serves mainly as a proof of concept that demonstrates the capacity of this framework to capture new interesting settings. Another contribution of our result for learning half-spaces is the kind of technical tools used in the construction (which are less commonly used in learning theory). See also our responses below to R4's comments. We also want to point out that the label-determined model does capture some realistic scenarios. For example, imagine a scenario where the data of individuals who tested positive for COVID-19 does not require privacy protection (to enable contact tracing and symptom analysis), while the data of those who tested negative remains protected. This is actually the case in some countries, and is exactly captured by the label-determined model.

- *R3: A short discussion on private knowledge transfer, and private learning halfspaces:* We will add a relevant discussion. Note that the prior works on private knowledge transfer (e.g., [Papernot et al. 2018, BTT18]) assume that public and private data are identically distributed. We also note that the target distribution in our case is a mixture of the (possibly different) private and public distributions. Hence, the knowledge transfer (or, domain adaptation) paradigm does not seem very useful in our case.

- *R4: "The extra novelty in the new privacy framework is that distributions of private and public data are different":* Our motivation for this new framework is to capture more realistic scenarios compared to prior work. One aspect, as the reviewer notes, is that the private and public data can arise from different distributions. Another aspect is that in this framework the examples are sampled from a single source (the mixture distribution) rather than assuming access to a separate oracle for each of the public and private examples (as in the prior work). So, unlike the prior work that utilized public data, here there is only one sample complexity, and it is given in terms of the sum of private and public examples drawn from the mixture.

- *R4: "I wonder what kind of results are even possible without making such assumptions":* There are various directions one can explore in this framework. Perhaps, among the most realistic ones are those based on a distribution-dependent model, where one also restricts the data distribution (and not only the concept class), or privacy models where the privacy status can be correlated with the feature vector not just the target label. Also, in the distribution-independent setting – as the reviewer notes – one may ask whether every VC class can be learned in the label-determined setting. (In the other extreme, where the privacy status and the label are independent, previous works [BNS13, ABM19] showed that any VC class is learnable, with significant savings in sample complexity.) Moreover, one may also consider privacy models that interpolate between these two extremes (label-determined and label-independent), and explore the sample complexity in this spectrum.

- *R4: The algorithm is improper and also not poly-time:* We note that improperness of the algorithm is not necessarily a drawback (e.g., boosting algorithms are improper). However, we think the question of whether one can construct a proper algorithm for this problem is an interesting open question as we mention in the paper. Concerning computational efficiency, note that, even without privacy, agnostic PAC learning of halfspaces is known to be computationally hard. However, the question of computational efficiency in the realizable case is yet another interesting question. Since our work makes the first step in studying this new framework, we believe it's useful to first study the sample complexity of this problem without computational restrictions.

- *R4: "Is it possible to get a sample complexity close to $O(d)$?" :* This is indeed a very good question for future work, which we have been thinking about. We will definitely add a relevant discussion to the paper.

- *R4: Comparison for the sample complexity of halfspaces with that of prior work:* We will add this comparison to the paper.

[Meta-Review · NeurIPS 2020]

This is a nice paper and the reviewers agreed that the results are interesting and non-trivial. Please keep in mind the detailed comments when preparing your camera-ready.